# Back to the light, coevolution between vision and olfaction in the "Dark-flies" (*Drosophila melanogaster*)

Ismet Özer[1], Thomas Carle[1,2]*

**1** Institute of Neuroscience, Framlington place, Newcastle University, Newcastle upon Tyne, United Kingdom, **2** Department of Biology, Faculty of Science, Kyushu University, Fukuoka, Japan

* th.carle@gmail.com

**Citation:** Özer I, Carle T (2020) Back to the light, coevolution between vision and olfaction in the "Dark-flies" (*Drosophila melanogaster*). PLoS ONE 15(2): e0228939. https://doi.org/10.1371/journal.pone.0228939

**Data Availability Statement:** All relevant data are within the manuscript and its Supporting Information files.

**Funding:** This work was supported by the European Commission (H2020-MSCA-IF-2015-

## Abstract

Trade-off between vision and olfaction, the fact that investment in one correlates with decreased investment in the other, has been demonstrated by a wealth of comparative studies. However, there is still no empirical evidence suggesting how these two sensory systems coevolve, *i.e.* simultaneously or alternatively. The "Dark-flies" (*Drosophila melanogaster*) constitute a unique model to investigate such relation since they have been reared in the dark since 1954, approximately 60 years (~1500 generations). To observe how vision and olfaction evolve, populations of Dark-flies were reared in normal lighting conditions for 1 ($DF_{1G}$) and 65 ($DF_{65G}$) generations. We measured the sizes of the visual (optic lobes, OLs) and olfactory (antennal lobes, ALs) primary centres, as well as the rest of the brain, and compared the results with the original and its genetically most similar strain (Oregon flies). We found that, whereas the ALs decreased in size, the OLs (together with the brain) increased in size in the Dark-flies returned back to the light, both in the $DF_{1G}$ and $DF_{65G}$. These results experimentally show that trade-off between vision and olfaction occurs simultaneously, and suggests that there are possible genetic and epigenetic processes regulating the size of both optic and antennal lobes. Furthermore, although the Dark-flies were able to mate and survive in the dark with a reduced neural investment, individuals being returned to the light seem to have been selected with reinvestment in visual capabilities despite a potential higher energetic cost.

## Introduction

The world is complex, and animals have evolved a large array of sensory systems to make sense of it. Getting information is so important for survival and reproduction that, at first sight, we may think that developing additional sensory capabilities in one or several modalities leads brain evolution. Obviously, those individuals that get more pertinent information have a clear advantage for finding food and sexual partners as well as for avoiding dangers. But, it also has an energetic cost for the brain to develop and maintain neural tissues [1–3]. Therefore, to invest in one or several sensory modalities makes sense only if it provides individuals with a

706699). The funder had no role in study design, data collection and analysis, decision to publish, or preparation of the manuscript.

**Competing interests:** The authors have declared that no competing interests exist.

reproductive and/or survival advantage; otherwise it is simply costly without any benefit, and not evolutionary stable.

Dim light/absence of light is a good example of an environment to observe evolution of sensory systems under environmental constraints. Dim light/absence of light exerts a selective pressure not only on vision, but also on other sensory systems. Until now, a wealth of comparative studies has shown that cave dwelling/nocturnal species have evolved either by developing specialised eyes, or by reducing the size of their eyes (for reviews: [1,4–7]). Not only dim light/absence of light is responsible for reduced eyes, but also captivity as observed in fruit flies (*Drosophila melanogaster*) [8]. Although effects of captivity on other sensory systems has not been documented, cave dwelling/nocturnal species with reduced investment in vision have also evolved with higher investment on alternative sensory systems, such as olfaction (e.g. [9–11]) or mechanoreception (e.g. [12–14]) to get information about their environment and to compensate the lack of visual information.

These observations initiated a common thinking that has relatively recently emerged about 'trade-off' between sensory systems, the fact that investment in one sensory modality correlates with decreased investment in another sensory modality. Trade-off between vision and olfaction has been described in several species such as in primates firstly [15], fish [7], butterflies/moths [10,16] and in ants [11,17]. However, whilst numerous comparative studies between several species have shown such trade-off between vision and olfaction (e.g. [7,10,11,15–19]), there is still no evidence that individuals are selected with inversed investments for vision and olfaction.

Empirical evidence is necessary not only to confirm this hypothesis of "trade-off", but also to understand the mechanisms of change in the balance between sensory systems. Changes of sensory systems might occur simultaneously meaning that reduced investment in one sensory modality and increased investment in an alternative sensory system occur at the same time. However, they might also occur consecutively. In this case, reduced investment in one sensory modality appears before increased investment in an alternative sensory system. Order of changes is important since it may provide cues about the origin of these changes. For example, simultaneous changes may be the consequence of genetic/epigenetic factors that control investment in several sensory modalities concurrently whereas consecutive changes may reflect independent mechanisms. However, species lifespan makes difficult to observe such trade-off over generations.

The 'Dark-flies', a strain of fruit flies (*Drosophila melanogaster*) reared in the dark for more than 60 years [20], constitute a unique model to investigate how individuals are selected based on sensory investment and to show a trade-off between vision and olfaction over generations. In the present study, we initially measured morphological differences (body and eyes sizes) between the Dark-flies and its parental and most genetically related strain (Oregon flies) [21]. To observe sensory investment in vision and olfaction over generations, a population of Dark-flies has been reared in normal lighting conditions for 65 generations ($DF_{65G}$) at the time of this study. We then measured the sizes of visual (optic lobes) and olfactory (antennal lobes) primary centres, as well as the size of the brain in the Dark-flies, $DF_{65G}$ and Oregon flies. To ensure that the differences observed between the Dark-flies and $DF_{65G}$ were not due to rearing conditions, we also measured the same parameters in a population of Dark-flies that has been reared in normal lighting conditions for 1 generation. Overall, we expected to observe larger optic lobes (OLs) in the $DF_{1G}$ as a consequence of the presence of light on the development of visual system, and no change concerning the antennal lobes (ALs). Concerning the $DF_{65G}$, we expected to observe that investment in vision increased beyond this developmental change (i.e. larger OLs compared to the $DF_{1G}$), and to observe a reduced investment in olfaction in the $DF_{65G}$ (i.e. smaller ALs).

## Methods

### Subjects and housing

In 1954, the Dark Flies Project was started by putting and maintaining in the dark populations of fruit fly (*Drosophila melanogaster*) taken from an original Oregon-R-S strain [20]. During the Dark-flies Project, the Dark-flies were maintained in sterilized milk-bottles plugged with a cotton-ball or silicon plug, and provided with a low-nutrient food source, Pearl's synthetic medium [22], to accentuate selection of individuals. The flies were kept in the dark by placing the bottles in a light-proofed can that was painted black inside and had a blackout curtain to cover the lid. In parallel, the Oregon-R-S strain was maintained under 12:12 LD lighting conditions, and fed within a standard cornmeal medium. All flies were kept at 25˚C in a temperature-controlled room. In 2002, the original Oregon-R-S strain from which the Dark-flies had been taken was lost, and a new population of Oregon-R-S was re-obtained and established from the original source. This new population of Oregon-R-S flies is the most genetically similar strain to the Dark-flies of those that have been analysed [23], and were used for comparisons in our study.

In November 2014, a population of Dark-flies was placed in a 12:12 L:D condition, similarly to the Oregon flies (Fig 1). This new strain, the original Dark-flies and Oregon-R-S flies were obtained in February 2017 from the Dark Flies Project based at Tohoku University, Japan [20]. Upon arrival, we kept all flies under the same standardised conditions for 6 months (about 12 generations) before making any measurements and starting experiments. This was to help reduce any effects of their past rearing environment or nutritional status on our results. All flies were reared in vials (28.5 x 95mm) of K-resin (VWR International) containing a fresh mixed plain white drosophila medium (Blades Biological Ltd). They were kept in a room at 25 ±3˚C with a 12:12h L:D photoperiod (light phase: 0700–1900). The tubes containing the Dark-flies were maintained in a metal container surrounded by black tissue paper to keep out the light.

### Body measurements

At 4 days old, Oregon flies and Dark-flies that had been kept under their standard rearing conditions were collected to measure their mass, length (Oregon: 49 males, 67 females; Dark-flies: 24 males, 41 females; Fig 2). The flies were anesthetised by putting their tubes into ice. Photographs of the flies were taken against a piece of graph paper (for scale) using a camera fixed on a microscope (Brunel eyecam plus fixed to a BMDZ Brunel Microscopes Ltd). The length of each fly was measured using GIMP 2 (version 2.8.22), and the body mass was measured using a balance with a range of 0.01mg (Mettler AT261 Professional Analytical Balance, © Mettler-Toledo).

### Brain extraction and histology

After sacrificing the flies by immerging them in ethanol (100%), we extracted their brains in a Ringer solution (NaCl: 180mM; KCl: 6mM; $CaCl_2$: 3mM; $NaHCO_3$: 3mM; Ph: 7.3). We then followed the method used in our previous study, which is a variant of Stölck and Heinze's method [24]. On the day of extraction, we fixed the brains in a zinc-formaldehyde fixative solution ($ZnCl_2$: 18.4mM; NaCl: 135mM, sucrose: 35mM; 1% paraformaldehyde; pH 7.3) at room temperature overnight. On the second day, we rinsed the brains 8 x 20min in PBS (NaCl: 140mM; KCl: 2mM; Na2HPO4: 10mM; KH2PO4: 2mM; pH 7.3) and bleached them in a fresh solution of 10% hydrogen peroxide in 0.05M Tris-buffered saline solution (Tris-HCl: 0.05mM) for 6 hours. After bleaching, we again rinsed the brains with a Tris-HCl solution (3 x

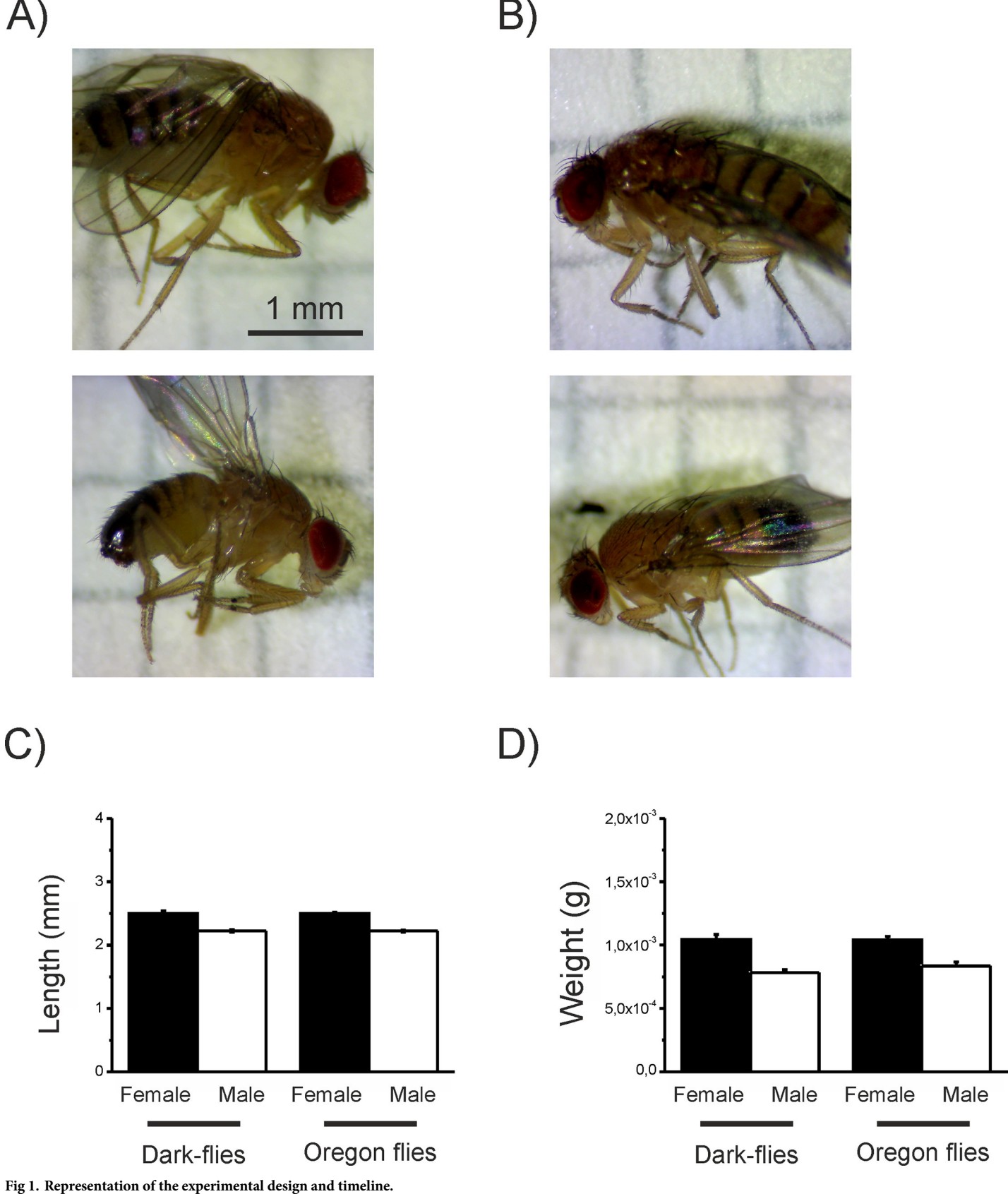

**Fig 1. Representation of the experimental design and timeline.**

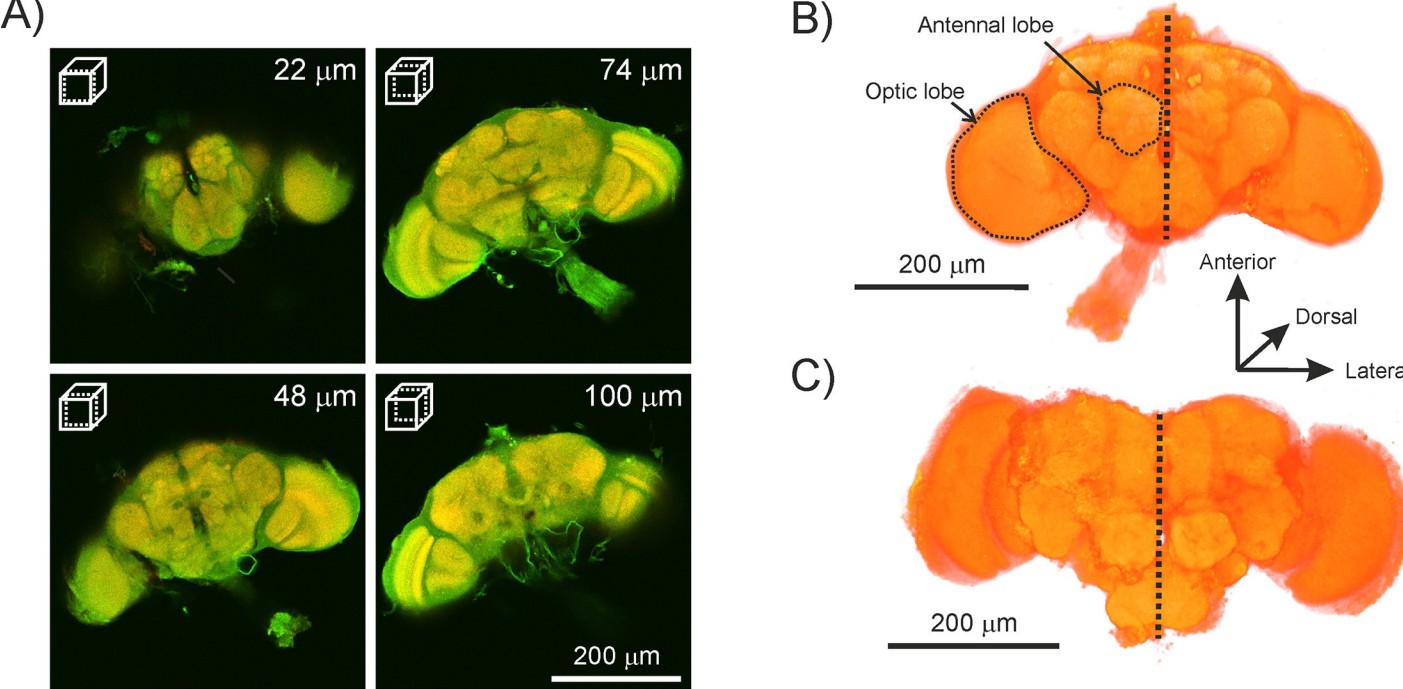

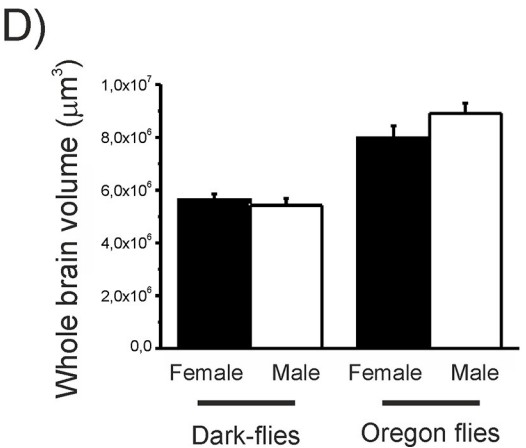

**Fig 2. Images of male (A) and female (B) flies from each strain.** In each image, Dark-flies are on the left, and Oregon flies on the right. The mean (+SEM) length (C) and mass (D) of male and female Oregon flies and Dark-flies.

10min), put them in a fresh mixture (20:80) of dimethyl sulfoxide (DMSO):Methanol for 85min, and rinsed a final time with a Tris-HCl solution (3 x 10min). After the last wash, we pre-incubated the brains overnight at 4°C in a solution of PBT (PBS with 0.3% of Triton X-100) containing 5% goat serum. On day 3, we incubated the brains at 4°C with 1:25 anti-synapsin antibodies (3C11 anti SYNORF1; Developmental Studies Hybridoma Bank; dshb.biology.uiowa.edu) added to PBT containing 1% goat serum for 5 days. On day 6, we washed the brains with a solution of PBT (8 x 20min), and then incubated them at 4°C with 1:200 of the secondary antibody (goat anti-mouse, IgG (H+L) conjugated to rhodamine; JacksonImmunoResearch, WestGrove, PA, USA) for 5 days. On day 13, we first again washed the brains

with a solution of PBT (2 x 30min), and then with a solution of PBS (6 x 30min). After these washes, we dehydrated the brains in an ascendant series of ethanol solutions (70%: 2 x 10min; 80%: 10min; 90%: 10min; 100%: 2 x 30min) before stocking them in methyl salicylate at -20°C.

### Confocal laser scanning microscopy (LSM)

We observed the stained brains using a confocal microscope (Nikon A1; Nikon corporation) with a X10 0.45-NA plan Apo λ objective. We used a helium-neon laser with a long-pass emission filter (561nm) to visualise the antibodies (anti-synapsin), and an argon laser with a band-pass emission filter (488nm) for background autofluorescence. We made optical sections at a resolution of 1,024 x 1,024 pixels with 2μm intervals through the entire depth of the brains following a ventral-dorsal neural axis [25]. In total, we obtained brains of 14 (7♂ and 7♀) Dark flies, 14 (7♂ and 7♀) $DF_{1G}$, 10 (5♂ and 5♀) $DF_{65G}$ and 10 (2♂ and 8♀) Oregon flies.

### Analyses and statistics

Body, eyes and brain measurements were made blind to fly strain: the optical image files were renamed with random numbers generated by Excel (Microsoft® Excel® for MAC 2011, version 14; © 2010 Microsoft Corporation) and were given to an observer who was unaware which strain each image was from. The volumes of the whole brain, the optic lobes (OLs), the antennal lobes (ALs) and the hemisphere (i.e. the whole brain without the OLs and ALs) were measured from the optical image files using FIJI software [26] and a plugin ("measure stack": Bob Dougherty, Copyright (c) 2002, 2005, OptiNav, Inc.). In the case of OLs, we measured the whole volume of optic lobes (i.e. medulla, lobula and lobula plate including neuropils and cell bodies) whereas the whole volume of glomeruli was considered for the antennal lobes. The relative volumes of OLs and ALs were calculated by dividing their volumes by the volume of hemisphere.

Statistical analyses were performed using SPSS v22 (IBM Corporation) using Generalized Linear Models (GLMs). For body and eye sizes, we performed GLMs to test the effects of fly strain (Dark-flies or Oregon flies) and sex (male or female) on the measurements. For brain measurements, we initially tested the effects of fly strain (Dark flies, $DF_{1G}$, $DF_{65G}$ or Oregon flies) and sex (male or female) on the data. Because we did not find any main effect of sex or interaction of sex with any other factor (see result section), we removed sex from the models, and performed GLMs with strain as a factor. Finally, we performed post-hoc tests with pairwise comparisons using Fisher's Least Significant Difference (LSD).

## Results

### Body and whole brain size

To explore whether there were any morphological and anatomical differences between the Dark-flies and the Oregon flies, we first compared the body and brain sizes between the two strains reared under their standard rearing conditions, i.e. Dark-flies reared in 24D and Oregon flies reared under a 12L:12D condition. There was no effect of strain on either body mass ($\chi^2_1 = 0.517$, P>0.05; Fig 2D) or length ($\chi^2_1 = 0.001$, P>0.05; Fig 2C), although females of both strains were consistently heavier (GLM; $\chi^2_1 = 45.53$, P<0.001; Fig 2D) and larger than males ($\chi^2_1 = 97.59$, P<0.001; Fig 2C), with no interaction between sex and strain for either measure (mass: $\chi^2_1 = 0.687$, P>0.05; length: $\chi^2_1 = 0.012$, P>0.05).

Although there were no significant differences between the strains in their body size, Dark-flies had smaller brains than Oregon flies (GLM: $\chi^2_1 = 64.48$, p<0.001; Fig 3D). In contrast to body size and mass, we did not find any effect of sex in the measurements ($\chi^2_1 = 0.99$, p>0.05).

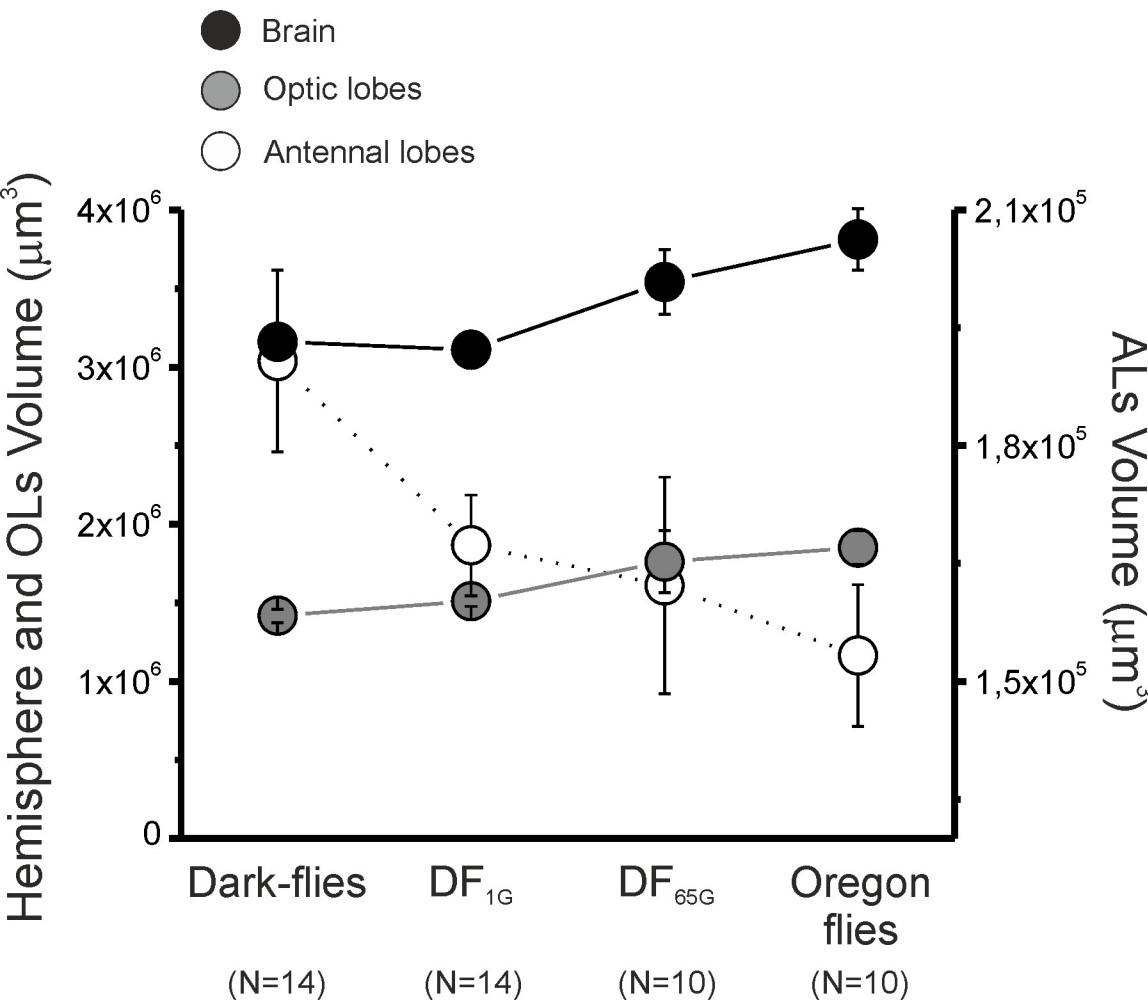

**Fig 3. Flies' brains. B**rain images from a brain of a male Dark-fly showing a series of optical sections from ventral (top left) to dorsal (bottom right) side following a neural axis (A). A 3D representation of a brain of a male Dark-fly (B) and a male Oregon fly (C). The scales and depth location are presented in the figure. The mean (+SEM) size of whole brain (including OLs and ALs) (D) in males and females Oregon and Dark-flies.

We also did not find any interaction between strain and sex ($\chi^2_1 = 2.54$, p>0.05) on whole brain measurements. Therefore, both male and female Dark-flies had consistently relatively smaller brains to body size than the Oregon flies.

### Absolute brain measurements

There was a main effect of the strain on the size of the hemisphere (GLM: $\chi^2_3 = 19.81$, p<0.001; Fig 4): the Dark-flies and $DF_{1G}$ had smaller hemispheres compared to the $DF_{65G}$ (p<0.05) and Oregon flies (p<0.001). Interestingly, there were no differences between the $DF_{1G}$ and the original Dark-flies (p>0.05), or between the $DF_{65G}$ and the Oregon flies (p>0.05).

The volume of the OLs (GLM: $\chi^2_3 = 13.94$, p<0.01) and ALs (GLM: $\chi^2_3 = 8.13$, p<0.05) also changed according to the strains (Table 1). The Dark-flies had smaller optic lobes compared to the $DF_{65G}$ (p = 0.011) and Oregon flies (p = 0.001) and, conversely, the Dark-flies possessed a larger absolute size of the ALs compared to the $DF_{65G}$ (p = 0.044) and Oregon flies

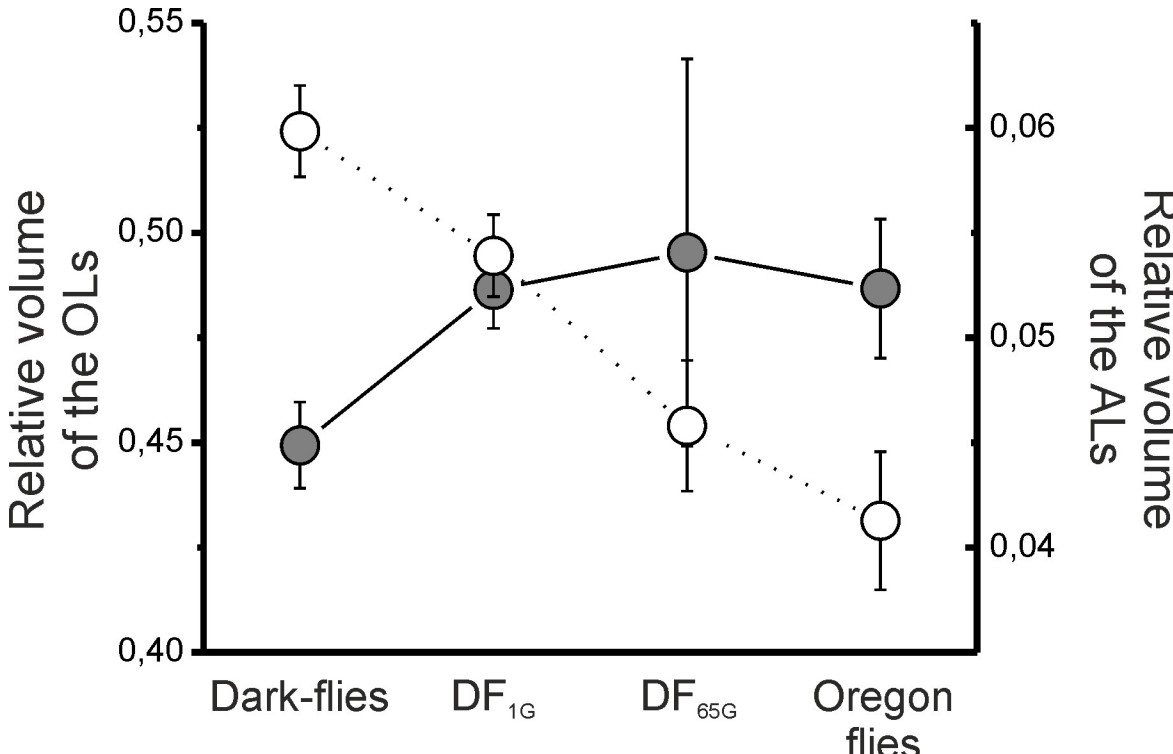

**Fig 4. Hemisphere, OLs and ALs absolute sizes.** Absolute volumes ± SEMs of hemisphere (black), OLs (grey) and ALs (white) in Dark-flies, $DF_{1G}$, $DF_{65G}$ and Oregon flies. Statistics are represented with letters associated to dots, and significant differences ($p<0.05$) are represented with different letters.

($p = 0.008$). Although there was no significant difference between the $DF_{1G}$ and $DF_{65G}$ flies ($p = 0.061$), the size of the optic lobes in the $DF_{1G}$ was significantly smaller than that of the Oregon flies ($p = 0.011$). For the ALs, there were no significant differences between the $DF_{1G}$, $DF_{65G}$ and Oregon flies (pairwise comparisons, for all values: $p>0.322$).

### Relative brain sizes

Concerning the relative sizes, there was an effect of the strains on the relative sizes of the ALs (GLM: $\chi^2_3 = 34.14$, $p<0.001$; Fig 5), but not on that of the OLs (GLM: $\chi^2_3 = 3.1$, $p = 0.376$). The Dark-flies and $DF_{1G}$ possessed larger relative ALs compared to the $DF_{65G}$ ($p<0.05$) and Oregon flies ($p<0.001$); however, no differences were observed between the Dark-flies and the $DF_{1G}$ ($p = 0.062$) or between the $DF_{65G}$ ($p = 0.28$) and the Oregon flies.

To confirm a tendency of a trade-off between vision and olfaction, we performed GLMs and tested whether the relative size of the OLs co-varied with the relative size of the ALs. By

**Table 1. Absolute and relative sizes of brain, OLs and ALs.**

| | Hemisphere (µm³) | | OLs size (µm³) | | ALs size (µm³) | | OLs relative size | | Als relative size | |
|---|---|---|---|---|---|---|---|---|---|---|
| | Mean | SEM | Mean | SEM | Mean | SEM | Mean | SEM | Mean | SEM |
| Dark-flies | **3,16E+06** | 7,96E+04 | **1,42E+06** | 4,33E+04 | **1,91E+05** | 1,16E+04 | **0,45** | 0,01 | **0,060** | 0,002 |
| $DF_{1G}$ | **3,11E+06** | 6,55E+04 | **1,51E+06** | 3,33E+04 | **1,67E+05** | 6,38E+03 | **0,49** | 0,01 | **0,054** | 0,002 |
| $DF_{65G}$ | **3,54E+06** | 2,05E+05 | **1,76E+06** | 1,97E+05 | **1,62E+05** | 1,38E+04 | **0,50** | 0,05 | **0,046** | 0,003 |
| Oregon flies | **3,81E+06** | 1,97E+05 | **1,85E+06** | 1,07E+05 | **1,53E+05** | 9,00E+03 | **0,49** | 0,02 | **0,041** | 0,003 |

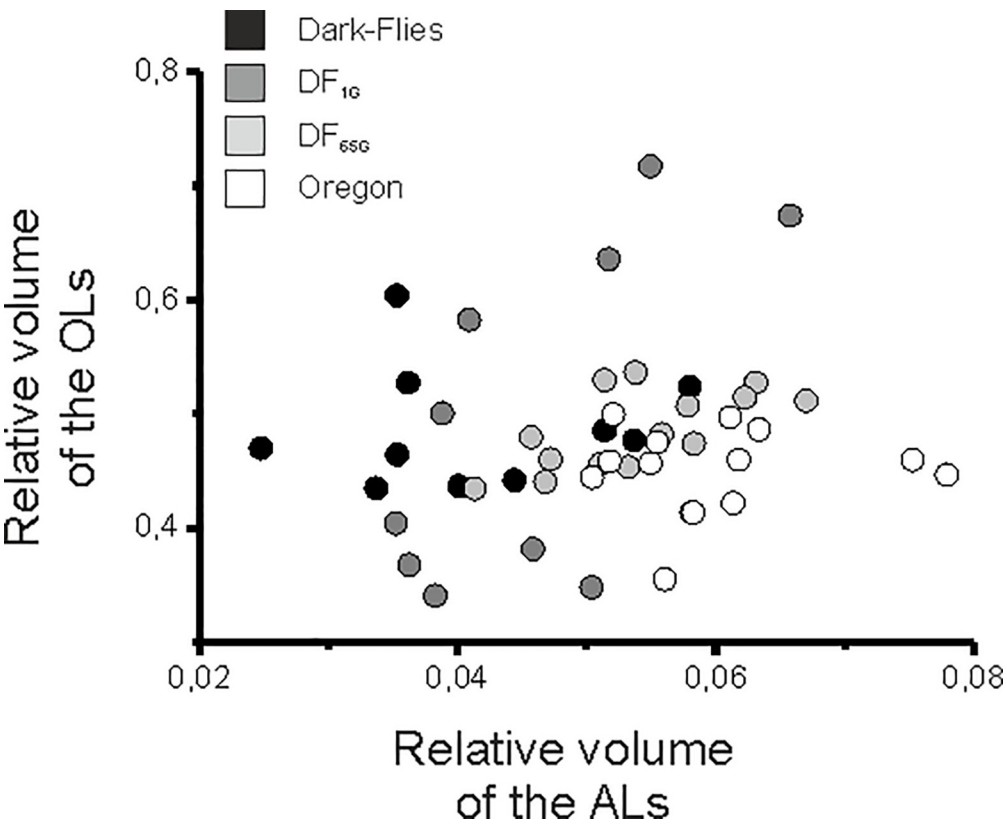

**Fig 5. OLs and ALs relative sizes.** Mean values of the relative sizes ± SEMs of OLs (grey) and ALs (white) in Dark-flies, $DF_{1G}$, $DF_{65G}$ and Oregon flies. The number of individuals used is shown in Fig 4. Statistics are represented with letters associated to dots, and significant differences (p<0.05) are represented with different letters.

performing such statistics, we found a major effect of the strains on the ALs (GLM: $\chi^2_3$ = 46.00, p<0.001), on the OLs (GLM: $\chi^2_3$ = 10.49, p<0.05), and a significant effect of the covariate (GLM: $\chi^2_1$ = 8.39, p<0.01).

## Discussion

Returned to the light, the Dark-flies, whose sensory investment seems in favour of olfaction, have clearly been selected with a decreased investment in the antennal lobes and increased investment in the optic lobes and whole brain in general, beyond developmental processes. This constitutes the first experimental evidence of a trade-off between vision and olfaction. This trade-off seems simultaneous at first sight, since we observed changes in both sensory systems in the $DF_{1G}$ and $DF_{65G}$, and might be associated to possible genetic or/and epigenetic processes regulating the size of both optic and antennal lobes. Furthermore, our results highlight the fact that darkness has a negative influence on brain size, which might be the consequence of a decreased investment in the visual system. In addition, after having adopted less energetically costly strategies for survival and reproduction in the dark, individuals have been selected with higher investment in the visual system after being returned to the light despite its potential energetic cost. Since there were no predatory or other environmental pressures (other than light condition) in our flies' raising conditions, selection of the flies might have been the result of sexual selection and less probably of foraging. In other words, it is unlikely that individuals possessed an advantage in terms of survival since food was really easy to find

in the tubes. However, it is likely that individuals possessing better visual capabilities had reproductive skills (e.g. better access to females during male competition) and inherit this trait on the following generations.

Our results experimentally show for the first time that the trade-off between vision and olfaction occurs simultaneously. The Dark-flies returned to the light for 1 and 65 generations possess bigger OLs associated with smaller ALs compared to the original Dark-flies (although the results were not significant in the case of the $DF_{1G}$). We still do not know if changes in a single sensory modality (*i.e.* only reduced investment in obsolete sensory systems or only increased investment in more reliable sensory systems) may evolve or not. But, although we do not exclude the possibility that sensory systems may evolve independently, our results show mechanisms regulating both vision and olfaction at the same time in accordance to a new environment, in terms of developmental (i.e. rearing environment of individuals) and evolutionary processes (increased changes through generations).

Gradual changes were observed between the Dark-flies, $DF_{1G}$ and $DF_{65G}$ tend to show that both developmental and evolutionary processes affect the size of hemisphere and sensory systems and play a role in the trade-off between vision and olfaction. Although differences measured between the dark-flies and $DF_{1G}$ were not significant, we do not exclude the hypothesis that light had a positive effect on the size of hemisphere and OLs, and a negative effect on the size of ALs in the $DF_{1G}$. Indeed, it is already known that rearing flies in the dark for one generation has a negative impact on the size of mushroom bodies (neuropils present in the hemisphere) and OLs [27,28]. In our experiment, the $DF_{1G}$ were made to simply observe the reversed mechanism and to distinguish between changes due to developmental processes and changes due to due to evolutionary processes. By doing it, we have clearly observed that the size of hemisphere and the relative size of ALs significantly differed between the $DF_{65G}$ and $DF_{1G}$. More generally, the differences observed between the Dark-flies and $DF_{1G}$ were accentuated between the dark-flies and the $DF_{65G}$. Indeed, although these differences were not significant between the $DF_{1G}$ and the Dark-flies, they were significant between the $DF_{65G}$ and the Dark-flies for all the parameters measured (except for the relative size of OLs), reflecting a process in two steps.

We could not determine the origin of mechanisms responsible for changes in the size of OLs and ALs, but they might correspond to a competition between axonal terminations coming from different sensory systems and/or to genetic/epigenetic processes. Indeed, one explanation may lie in the neurons at higher levels that might have limitations in the number of dendritic connections that they can form. In such case, these connections would be favoured for one sensory modality to the detriment of another sensory modality in accordance with the environment (see [28]). It is likely that this process is involved in developmental differences. In particular, the Hox genes or the allele Lobe ($L^1$) could be at the origin of such explanation. It is well known that the Hox genes are involved in establishing regionalized identity of neurons in the embryonic brain and control the termination of neuronal proliferation by posteriorly inducing apoptotic cell death during postembryonic brain development [29]. More recently, the allele L1 has been showed to have an antagonistic action in the number of trichoid sensilla (on antennae) and the number of ommatidia (on eyes) in fruit flies [19].

However, another explanation might be the control of the size of both visual and olfactory systems by genetic and epigenetic processes that act anteriorly to develop one sensory system to the detriment to the other and might correspond to evolutionary changes. In particular, both visual and olfactory systems are developed from the differentiation of cells from the eye-antennal imaginal disc in fruit flies [30]. It has recently been shown that this differentiation is directly controlled by the gene Paired box 6 (Pax6) [31], and that this gene (Eyeless/Pax6) is involved in the trade-off between vision and olfaction [32]. In the present case, we still do not

know if this gene is responsible for the differences between the different strains, however, we may wonder if this would be equally applied to species where the olfactory and visual neuroepithelia develop from other and less competitive processes.

Our data tend to show that sensory systems are not equally weighted in terms of magnitudes concerning changes in neural substrates. We do not know if, other than vision and olfaction, any change took place in other sensory modalities such as the mechanosensory system. However, to compensate a decrease of $\sim 0.3 \times 10^5$ $\mu m^3$ in terms of neural substrate in the antennal lobes, the $DF_{65G}$ have developed $\sim 0.34 \times 10^6$ $\mu m^3$ of neural substrate in the optic lobes in association with $\sim 0.4 \times 10^6$ $\mu m^3$ in the hemisphere (Fig 3, Table 1). These changes in neural substrates seem directly associated with higher food consumption [33]. These results might indicate that the energetic cost of information is relative to sensory systems. For example, visual information seems more energetically costly than olfactory information in terms of neural substrates since the development of OLs is ten times higher than the reduction of ALs. However, the development of visual system seems to be relatively less costly in terms of evolutionary benefits that it provides for survival and reproduction.

This leads to consider that the balance between the visual and olfactory systems reaches an equilibrium where information is optimised to benefit individuals according to environment. It is interesting to notice that the brain areas measured in the $DF_{65G}$ tend to reach those of the Oregon flies. But, would these values reach those of the Dark-flies if these $DF_{65G}$ or other flies are put in the dark? This raises other questions such as to know if the reduction in the visual system in the dark would have been so drastic if the flies had abundance of food, or if the presence or absence of scents in the environment plays a role in the evolution of olfaction in the dark, and consequently on the evolution of vision. Here, there is a large horizon for questions in order to understand the co-evolution of sensory systems, questions we do not have the answers.

In the present study, we did not measure the eyes shape, the size of ommatidia and did not count their number to determine intra- and inter-specific differences (see [34]). However, since we did not find any difference in the surface of the eyes between the Dark-flies and Oregon flies (see S1 Fig), our results are towards a contradictory standpoint from those made in crickets *Gryllus bimaculatus* showing that individuals reared in the dark present an increase of the surface of their eyes in parallel to an increase in the number of ommatidia [35]. Since the crickets were reared for a single generation in the dark, we do not know if this difference between crickets and fruit flies is attributed to differences between developmental adaptations and long term rearing in the dark. Another explanation may lie to different developmental processes between crickets and fruit flies, i.e. hemimetabolous vs holometabolous developmental processes, which may raise the same question as previously on differences between species having different developmental processes.

Whilst many comparative studies have put the spotlight on cognitive abilities being central to brain size evolution (e.g. [36–38] but [39,40]), our empirical approach demonstrates that changes in sensory input, specifically vision, can play a key role in determining absolute hemisphere size. This is perhaps not surprising, given that processing visual information takes up so much of the brain in the *Drosophila*: the optic lobes represent around 30% of the whole brain, and are an order of magnitude larger than the antennal lobes. Additionally, other species also show such correlation between brain size and the size of the visual system such as in guppies [41], birds [42] and primates [18]. Our data complete these studies and support the idea of concerted brain evolution, the fact that there are constraints that cause correlated size changes across different component areas [43]. In the present case, darkness rather than captivity (since both Dark-flies and Oregon flies were reared in the same conditions of captivity) had a negative effect on brain size that may be explained by the reduced investment in the visual

system, data that join those of Barth *et al.* on neural development in the dark in fruit flies (*Drosophila melanogaster*) [27,28].

No data to date have referred to any correlation or absence of correlation between brain size and the size of the olfactory system until now. The optic lobes being ten times larger than the antennal lobes may explain why we found no correlation between the sizes of ALs and brain, in the sense that size variations of OLs will involve the largest variations of brain size. These data support the idea that different areas of the brain can change size relative to one another (mozaic brain evolution: [44,45]), and that environmental constraints and changes in the investment in sensory systems are important in determining brain size, in agreement with recent studies [46–48].

Trade-off between sensory systems is not limited to vision and olfaction only, and investment in sensory systems depends not only on species' environment, but also on lifestyle. For example, Barton *et al.* [15,18] found that primates have evolved visual capabilities to the detriment of olfaction. However, although they found a negative correlation between these two sensory systems in primates, they also found a non-significant negative relationship in insectivores and no correlation in bats [15]. Obviously, species did not evolve in the same manner, and vary in their reliance on sensory systems. For example, it is well known that cavefish or fish living in the deep sea have not only developed capabilities in olfaction, but also in mechanoreception [7,14]. The naked mole-rat and star-nosed mole are other examples of species possessing tiny eyes that mostly rely on tactile cues [12,13]. Therefore, trade-offs/coevolution between sensory systems might concern the senses in general. In the case of the fruit flies, although this might not have affected the balance between vision and olfaction, mechanoreception might also play a role, notably because it was observed that the Dark-flies possess longer bristles to detect tactile information and vibrations [49]. One of the challenges for the future will be to integrate other senses in comparative studies, and to understand the coevolution between sensory systems by taking into account 1) the sensory specificities of each species; 2) the fact that the mechanisms of neural integration differ between sensory systems. This might make measures of brain areas not sufficient, and other alternatives might have to be found in the future for measuring sensory capabilities.

To conclude, our observations are based on artificial selection in fruit flies in accordance with their lifestyle, but other observations might be made on different species/environments. Here, we have brought new highlights about the coevolution between vision and olfaction in the light of darkness, but also many new questions. These questions are not only essential to understanding how species have evolved in the past, but are oriented to anticipate the future of species. For example, it has been shown recently that human activity pushes animals to adopt a more nocturnal lifestyle [50]. Therefore, it becomes important to understand how individuals are selected and will perceive the world in the dark, and whether our impact on species' shift to nocturnal life is reversible, since we are at the origin of such evolutionary change. However, human activities are not only pushing species to more nocturnal environments, but are, in general, changing the lifestyle and evolutionary pressures on species, which may open the way to new studies.

## Ethics

The work was with insects, which are not subject to the same legal and regulatory rules as vertebrates, and do not fall under EU Directive 2010/63/EU on the protection of animals used for scientific purposes. However, a general ethical approval for the project was granted by Newcastle University for this project.

## Supporting information

**S1 Fig. Images of the eyes of Dark-flies (A) and Oregon flies (B) in females (top) and males (bottom).** The mean (+SEM) surface of a single eye (c) of male and female Dark-flies and Oregon flies.
(TIF)

**S1 File. Eyes measurement's method and results.**
(PDF)

## Acknowledgments

We would like to really thank Prof. Candy Rowe for her valuable support and warm welcome in her laboratory, Prof. Shuiti Mori for initiating this project in 1954 and Dr Naoyuki Fuse for providing us with Dark and Oregon flies.

## Author Contributions

**Conceptualization:** Thomas Carle.

**Data curation:** Ismet Özer.

**Formal analysis:** Ismet Özer.

**Funding acquisition:** Thomas Carle.

**Investigation:** Thomas Carle.

**Methodology:** Thomas Carle.

**Project administration:** Thomas Carle.

**Resources:** Thomas Carle.

**Supervision:** Thomas Carle.

**Writing – original draft:** Thomas Carle.

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
