## [Decision Letter · Decision Letter 0]

4 Nov 2019

PONE-D-19-27069

Back to the light, coevolution between vision and olfaction in the “Dark-flies” (Drosophila melanogaster)

PLOS ONE

Dear Dr Carle,

Thank you for submitting your manuscript to PLOS ONE. After careful consideration, we feel that it has merit but does not fully meet PLOS ONE’s publication criteria as it currently stands. Therefore, we invite you to submit a revised version of the manuscript that addresses the list of concerns raised during the two external reviewers.

We would appreciate receiving your revised manuscript by Dec 19 2019 11:59PM. To enhance the reproducibility of your results, we recommend that if applicable you deposit your laboratory protocols in protocols.io, where a protocol can be assigned its own identifier (DOI) such that it can be cited independently in the future. For instructions see: http://journals.plos.org/plosone/s/submission-guidelines#loc-laboratory-protocols

We look forward to receiving your revised manuscript.

Kind regards,

Matthieu Louis

Academic Editor

PLOS ONE

Journal Requirements:

1. Please ensure that you refer to Figure 2 in your text as, if accepted, production will need this reference to link the reader to the figure.

<h3>**2. Please upload a copy of Figure 5, to which you refer in your text on page 11. If the figure is no longer to be included as part of the submission please remove all reference to it within the text.**</h3>

3. Thank you for including your competing interests statement; "No"

Reviewers' comments:

Reviewer's Responses to Questions

**Comments to the Author**

1. Is the manuscript technically sound, and do the data support the conclusions?

Reviewer #1: Partly

Reviewer #2: Partly

2. Has the statistical analysis been performed appropriately and rigorously? 

Reviewer #1: Yes

Reviewer #2: Yes

3. Have the authors made all data underlying the findings in their manuscript fully available?

Reviewer #1: No

Reviewer #2: Yes

4. Is the manuscript presented in an intelligible fashion and written in standard English?

Reviewer #1: Yes

Reviewer #2: No

5. Review Comments to the Author

Reviewer #1: Impacting environmental changes provide a core driving force for evolutionary adaptations. How the nervous system is remodelled to cope with such changes remains largely unknown. The rapid generation time of fruit flies allows to investigate the genetic basis of adaptations.

In the work described in the current manuscript aims to address changes in brain architecture in flies kept in constant darkness for generations. The authors provide evidence that there are adaptations in the size of the antennal lobe, optic ganglia and overall brain size.

While the overall motivation for the study is indeed of interest there are several points that remain unclear.

- It is entirely unclear what flies were used for which type of analysis. In the method section the authors try to elaborate on the history of the “dark flies”, however it remains unclear if the original dark-fly stock was lost or if only the control stock was lost. Also, how this relates to the flies from 2014 is unclear.

It is critical that the authors clearly describe how the current experiment was done.

The authors may choose to add a graphical representation of the actual experiments performed.

- In the first result section “Body, eye and whole brain size” the eye-size analysis seems to have been forgotten!

While I assume that there was no difference (based on the supplemental data), there are a couple points taken into account.

1- Since the eye is an uneven surface the measurement may not be accurate. This may be improved by taking a high-resolution image of the eye in the focal plane (or scanning electron microscopy).

2- While not absolutely critical, this notion raises the question if the number of ommatidia is decreased.

- The volumetric analysis, a main point in the current manuscript remains poorly explained. The authors used the “measure stack” plugin in FIJI, however it is not described how different brain areas were distinguished and if this was based on the entire optic lobe ganglia (including cell bodies/cortex) or only neuropil (which I would assume was done for the antennal lobe neuropil).

- I feel it is critical to note that simply by exposing the animals to light it is unclear if and what type of selection occurred. Since flies are raised in rather artificial laboratory conditions and no controlled selective pressure was applied it is unclear what the relevant changes reflect. This should be clearly defined and discussed.

- The data displayed in Figure 3 are surprising. While it seems that the overall brain volume and optic lobe volume increases from Dark-flies to DF1G to DF65G and OR flies, the main change in antennal lobe size appears to occur from Dark-flies to DF1G flies. How can this be explained? Such large adaptation occurring in one generation seems rather strange.

Wording:

130 “After killing the flies “ should be “Flies were scarified” or similar.

Reviewer #2: 1. Due to lack of clarity in several parts of the manuscript it is hard to be fully confident that all conclusions are sound (see the review document).

4. As mentioned above, the manuscript needs to be revised for clarity.

6. PLOS authors have the option to publish the peer review history of their article (what does this mean?). If published, this will include your full peer review and any attached files.

Reviewer #1: No

Reviewer #2: Yes: Primoz Ravbar

---

## [Author Response · Author response to Decision Letter 0]

13 Dec 2019

Reviewer #1

- It is entirely unclear what flies were used for which type of analysis. In the method section the authors try to elaborate on the history of the “dark flies”, however it remains unclear if the original dark-fly stock was lost or if only the control stock was lost. Also, how this relates to the flies from 2014 is unclear. It is critical that the authors clearly describe how the current experiment was done. The authors may choose to add a graphical representation of the actual experiments performed.

We thank the reviewer for this remark. We agree that a graphical representation is better to understand the evolution of flies, and have included a new figure (Figure 1) in order to explain it.

- In the first result section “Body, eye and whole brain size” the eye-size analysis seems to have been forgotten!

While I assume that there was no difference (based on the supplemental data), there are a couple points taken into account.

1- Since the eye is an uneven surface the measurement may not be accurate. This may be improved by taking a high-resolution image of the eye in the focal plane (or scanning electron microscopy).

2- 2- While not absolutely critical, this notion raises the question if the number of ommatidia is decreased.

Since our analysis for the eye-size is far for being complete, we decided to keep this analysis as supplemental data in order not to weaken this article. The questions of the reviewer are very interesting. However, we are not able to make images with a better resolution since we are not located in Newcastle anymore. This also justified our choice to keep these data as supplemental data. 

- The volumetric analysis, a main point in the current manuscript remains poorly explained. The authors used the “measure stack” plugin in FIJI, however it is not described how different brain areas were distinguished and if this was based on the entire optic lobe ganglia (including cell bodies/cortex) or only neuropil (which I would assume was done for the antennal lobe neuropil).

We added details within the text. 

- I feel it is critical to note that simply by exposing the animals to light it is unclear if and what type of selection occurred. Since flies are raised in rather artificial laboratory conditions and no controlled selective pressure was applied it is unclear what the relevant changes reflect. This should be clearly defined and discussed.

We appreciate this comment and we added a part at the beginning of the discussion about it (Lines 371-377).

- The data displayed in Figure 3 are surprising. While it seems that the overall brain volume and optic lobe volume increases from Dark-flies to DF1G to DF65G and OR flies, the main change in antennal lobe size appears to occur from Dark-flies to DF1G flies. How can this be explained? Such large adaptation occurring in one generation seems rather strange.

We thank to the reviewer for this substantial insight. We do not have any explanation about the fact that the ALs decreased in size mostly between the Dark-flies and the DF1G. Since, we do not have any explanation about it, we were unable to discuss about it. If the reviewer would have some idea about such phenomenon, we are ready to make changes accordingly. 

Wording:

130 “After killing the flies “ should be “Flies were scarified” or similar.

The line has been changed accordingly. 

Reviewer #2

- Overall the paper is interesting, the methods, results and the conclusions seem generally credible. However, the manuscript needs major revisions in terms of clarity. In the review below only several specific cases where the revisions is necessary are stated, however, the manuscript probably needs major revisions way beyond these examples. In particular the hypothesis and predictions should be clearly stated. The Figures and the Results should address the predictions clearly and any discrepancies between the predictions and the results should be pointed out and thoroughly discussed. The justifications for the control groups used should also be made more clear. 

- This reviewer recommends the manuscript for publication based on intriguing, indeed fascinating, question of how development and evolution of sensory systems can be affected by depriving flies of one sensory modality (vision) across many generations, in a controlled environment. The reviewer is confident that the authors will be able to present their current results and conclusions in a clear form in the revised versions. 

Line 14: “the Dark-flies returned back to the light” Does this refer to DF65G flies? 

The line refers both to DF1G and DF65G. Appropriate clarifications are made. 

Line 36: “environment to see evolution” Maybe: “… to observe evolution...”

The line has been changed accordingly. 

Line 54: “between extent species” Maybe: “Between several species”?

We kept the term extent in the sense that we would like to refer to species that are still present nowadays on the earth, in opposition to fossils and other species that disappeared. Since analyses may be made on fossils, we would like to keep this distinction and hope that this is suitable for the reviewer. 

Line 64: “...or alternative...” Do the authors mean “consecutive”? This paragraph should be re-written for clarity. In particular lines 66-68 are hard to comprehend by this reviewer. 

We thank the reviewer for this suggestion. The whole paragraph has been adjusted in order to be more unambiguous. We also kept the work « consecutive » as suggested. 

Lines 82-88: Another control for DF65G flies should be Dark-flies treated exactly the same as the experimental group but without light. If this is what was done, authors should clarify. It would be nice to observe any dosage effect between flies raised in light for various numbers of generations. 

Although the suggestion makes sense and we agree that the DF65G without light would be interesting to observe, unfortunately we do not have the appropriate data and cannot perform this experiment anymore. We really thank the reviewer for this suggestion and this would be an interesting point to investigate in the future, although Barth et al. investigated brain and OLs sizes in fuit flies raised in the dark for one generation. 

There are two possible mechanisms for the sensory trade-off that the authors observe. One is a per-existing stimulus-dependent plasticity, whereby brain development of these sensory areas is a function of sensory input (activity-dependent plasticity). The other mechanism for the trade-off would be “hard-wired” changes of the sensory areas resulting from evolutionary selection. If the former mechanism is the case, we should expect to observe the differences in sensory areas between Dark-flies and DF1G, while in the case of the latter mechanism, the evolutionary selection, the differences should only be observed between Dark-flies (or DF1G flies) and the DF65G flies. The authors should clarify their motivation for setting the control groups.

We thank to the reviewer for this substantial insight. We included a new paragraph in the discussion to make things less confusing (lines 413-430). We hope that this paragraph may satisfy the reviewer and reply to questions that were unclear until now. 

Line 98: the line should read: “...12:12 LD lighting conditions...”

Appropriate changes have been made.

Lines 194-198: authors should state the N (number of flies) for each group across the comparisons. 

The number of flies that were used for each experiment was already included in the material and method section. In order to make things clearer, we added these numbers on the figures. We hope that this change is adequate for the reviewers. 

Line 202: the text is probably referring to Fig. 2D, not Fig 3D .

Appropriate changes have been made. More generally, our references to figures were erroned and we apologize for such mistake. 

Lines 208-214: Should the text be referring to Fig. 3 rather than Fig. 4? If so, then it makes sense. Otherwise it is difficult to see how these results are reflected in the figure. 

Appropriate changes have been made.

Lines 224-232: the text is probably referring to Fig. 4, not Fig 5. The authors state that there was no effect on OL size in Fig. 4 yet there seems to be a significant difference between the Dark-flies and the other three categories in terms of OL size.

Appropriate changes have been made.

Line 235 (and elsewhere): should “main effect” be stated as “major effect”? 

Appropriate changes have been made.

Lines 258-260: If the difference in OL and AL sizes were significant between Dark-flies and DF1G flies, wouldn’t this suggest a developmental rather than evolutionary cause? Are the authors suggesting that a combination of developmental and evolutionary mechanisms is implicated in the observed differences? Please clarify.

We appreciate the comments. Indeed, this is the case. We added a paragraph in order to make the discussion clearer (lines 413-430) and included this way of thinking by talking about a process in two steps.

Line 295: “… i.e. mechanosensory system ...” “e.g.”?

Appropriate changes have been made.

Lines 292-304: this paragraph is difficult understand and the references to the figure and the table are unclear. Should be re-written. 

We modified this paragraph in order to make it more comprehensible. We hope that the changes that were made are adequate and made it understandable. 

Line 380: “the light of darkness” should read “the light or darkness”

We kept the terms « light of darkness » since our wish was to make a word play here, in order words, we would like to express the fact that darkness (the fact that we used darkness) enabled us to bring new insights. 

Figure 1: The number of flies in each group should be stated in the figure.

For the figures, please note that the number of the figures changed since we added one more figure. For the previous figure 1, newly figure 2, we added the number of flies as requested. 

Figure 2: Label the fly groups in the figure. 

We have labeled the brain with the group. Concerning the bar plot, the groups are indicated. If there is something else that we missed for changing the figure and in accordance with the reviewer, we are open for making changes. 

Figure 3: Add the number of flies in each category. Add p-values or other indicators of significance to the figure. Confidence intervals are hard to read for DF65G group for (OL and AL).

The number of flies is already present on the figure (on the bottom). If the reviewer could be more explicit, we can make changes according to his request. We changed the confidence interval to make it clearer and added letters to represent significant changes. We hope that these changes are adequate for the reviewer. 

Figure 4: Add the legend. Units on y-axis are not defined. Confidence intervals are hard to read for DF651 group for (OL and AL).

We added the legend and changed the confidence intervals in order to make it more clear. Concerning the X-axis, we added some marks to make it clearer.

---

## [Decision Letter · Decision Letter 1]

28 Jan 2020

Back to the light, coevolution between vision and olfaction in the “Dark-flies” (Drosophila melanogaster)

PONE-D-19-27069R1

Dear Dr. Carle,

We are pleased to inform you that your manuscript has been judged scientifically suitable for publication and will be formally accepted for publication once it complies with all outstanding technical requirements.

With kind regards,

Matthieu Louis

Academic Editor

PLOS ONE

Additional Editor Comments (optional):

Reviewers' comments:

Reviewer's Responses to Questions

**Comments to the Author**

1. If the authors have adequately addressed your comments raised in a previous round of review and you feel that this manuscript is now acceptable for publication, you may indicate that here to bypass the “Comments to the Author” section, enter your conflict of interest statement in the “Confidential to Editor” section, and submit your "Accept" recommendation.

Reviewer #1: All comments have been addressed

Reviewer #2: All comments have been addressed

2. Is the manuscript technically sound, and do the data support the conclusions?

Reviewer #1: Yes

Reviewer #2: Yes

3. Has the statistical analysis been performed appropriately and rigorously? 

Reviewer #1: I Don't Know

Reviewer #2: N/A

4. Have the authors made all data underlying the findings in their manuscript fully available?

Reviewer #1: No

Reviewer #2: (No Response)

5. Is the manuscript presented in an intelligible fashion and written in standard English?

Reviewer #1: Yes

Reviewer #2: Yes

6. Review Comments to the Author

Reviewer #1: In the revised manuscript the authors have removed the incomplete eye analysis, which makes the manuscript more coherent (Except in method section line 170 the eye is still included). Similarly most of the methodology is more appropriately represented and can be easily followed.

My only minor remaining concern lies in the surprising finding of the antennal lobe size of DF1G.

Reviewer #2: (No Response)

7. PLOS authors have the option to publish the peer review history of their article (what does this mean?). If published, this will include your full peer review and any attached files.

Reviewer #1: No

Reviewer #2: Yes: Primoz Ravbar

---

## [Editor Report · Acceptance letter]

4 Feb 2020

PONE-D-19-27069R1 

Back to the light, coevolution between vision and olfaction in the “Dark-flies” (Drosophila melanogaster) 

Dear Dr. Carle:

I am pleased to inform you that your manuscript has been deemed suitable for publication in PLOS ONE. Congratulations! Your manuscript is now with our production department. 

With kind regards,

on behalf of

Dr Matthieu Louis 

Academic Editor

PLOS ONE